# Hexanary blends: a strategy towards thermally stable organic photovoltaics

Sri Harish Kumar Paleti [1] ✉, Sandra Hultmark[2], Jianhua Han[1], Yuanfan Wen [1], Han Xu[1], Si Chen[1], Emmy Järsvall[2], Ishita Jalan [3], Diego Rosas Villalva [1], Anirudh Sharma [1], Jafar. I. Khan[1], Ellen Moons [4], Ruipeng Li[5], Liyang Yu[6], Julien Gorenflot [1], Frédéric Laquai [1], Christian Müller [2] ✉ & Derya Baran [1] ✉

Non-fullerene based organic solar cells display a high initial power conversion efficiency but continue to suffer from poor thermal stability, especially in case of devices with thick active layers. Mixing of five structurally similar acceptors with similar electron affinities, and blending with a donor polymer is explored, yielding devices with a power conversion efficiency of up to 17.6%. The hexanary device performance is unaffected by thermal annealing of the bulk-heterojunction active layer for at least 23 days at 130 °C in the dark and an inert atmosphere. Moreover, hexanary blends offer a high degree of thermal stability for an active layer thickness of up to 390 nm, which is advantageous for high-throughput processing of organic solar cells. Here, a generic strategy based on multi-component acceptor mixtures is presented that permits to considerably improve the thermal stability of non-fullerene based devices and thus paves the way for large-area organic solar cells.

The development of non-fullerene acceptors (NFAs) has resulted in a significant increase in the power conversion efficiency (PCE) of solution-processed organic solar cells (OSCs). Devices fabricated by lab-scale spin coating have reached a PCE > 17 %[1–3] while a PCE of 12.6%[4] has been reported for larger-area modules fabricated using high-throughput coating techniques. These highly promising results are typically obtained for as-prepared devices. However, the poor intrinsic stability of the active layer of NFA based organic solar cells under prolonged exposure to light and/or heat tends to result in a gradual decrease of the initial device performance, which stands in the way of full-scale commercialization[5–8].

The active layer of an OSC typically comprises a mixture of donor and acceptor molecules, a so-called bulk-heterojunction (BHJ), whose optimal nanostructure is highly material specific. The nanostructure of a BHJ blend comprises both, mixed regions where the donor and acceptor molecules are in intimate contact, as well as relatively pure domains of either blend component, which facilitate exciton dissociation and charge extraction, respectively. The poor intrinsic stability arises because the optimal nanostructure of a best performing BHJ tends to be far away from thermodynamic equilibrium[9]. As a result, the initial BHJ nanostructure can evolve with time through short- and/or long-range diffusion of either of the blend components resulting in a decrease in device performance[10,11].

On the one hand, long-range diffusion can result in the formation of crystals with sizes bigger than the exciton diffusion length, increasing geminate recombination in an OSC[12]. One prominent way to impede long-range mass transport is by realizing glassy BHJ blend films with a glass transition temperature $T_g$ far above their operating

[1]King Abdullah University of Science and Technology (KAUST), Physical Sciences and Engineering Division (PSE), KAUST Solar Center (KSC), Thuwal 2395 5-6900, Kingdom of Saudi Arabia. [2]Department of Chemistry and Chemical Engineering, Chalmers University of Technology, Göteborg 41296, Sweden. [3]Department of Engineering and Chemical Sciences, Karlstad University, Karlstad 65188, Sweden. [4]Department of Engineering and Physics, Karlstad University, Karlstad 65188, Sweden. [5]National Synchrotron Light Source II, Brookhaven National Lab, Upton, NY 11973, USA. [6]School of Chemical Engineering, College of Chemistry and State Key Laboratory of Polymer Materials Engineering, Sichuan University, Chengdu 610065, P. R. China. ✉e-mail: paleti@chalmers.se; christian.muller@chalmers.se; derya.baran@kaust.edu.sa

conditions[9,13,14]. Fortunately, non-fullerene acceptors like ITIC, Y6 and their derivatives feature a high $T_g$ of about 180 °C[10,15] and 205 °C[16], respectively (see Fig. 1 and Supplementary Fig. 1 for chemical structures), and are thus less prone to long-range diffusion. On the other hand, short-range diffusion can lead to a variety of scenarios, including diffusion-limited crystallization[15] or purification of phases even at temperatures $T \ll T_g$[17], affecting charge recombination and extraction of aged devices. For example, Ghashemi et al., have shown that the acceptor Y6 in binary blends with the benzodithiophene based copolymer PBDB-T (see Fig. 1 and Supplementary Fig. 1 for chemical structures) undergoes a non-negligible rate of diffusion even far below its $T_g$[18]. Similarly, in blends of Y6 and the donor polymer PM6 (PBDB-T-2F; see Fig. 1 for chemical structure) the acceptor can diffuse out of the donor-rich mixed phase leading to over purification of the donor phase due to the low miscibility of Y6 in PM6[17]. Evidently, a high $T_g$ alone is not sufficient to arrest the gradual evolution of the BHJ nanostructure of blends comprising non-fullerene acceptors.

One strategy that allows to further improve the thermal stability of non-fullerene based organic solar cells is the use of $D:A_1:A_2$ ternary blends. Some acceptor mixtures display a significantly reduced tendency toward acceptor crystallization as a result of the increase in entropy upon mixing, which reduces the driving force for crystallization[9]. For example, NFAs such as those based on indacenodithiophene and dihydroindenofluorene cores (e.g. IDTBR and IDFBR; see Supplementary Fig. 1 for chemical structures) can form a mixed phase[19,20]. Other acceptor mixtures such as ITIC-4Cl and ITIC-4F (see Supplementary Fig. 1 for chemical structures) undergo co-crystallization driven by favorable halogen interactions between the two acceptors, which is however suppressed once blended with the donor polymer PTB7-Th (see Supplementary Fig. 1 for chemical structure), resulting in improved device stability compared to devices based on binary blends[10]. In another study, it was shown that the thermal stability of $D:A_1:A_2$ ternary blends depends on the film thickness[21]. Yang et al. have reported that devices based on binary and ternary blends of Y6 and PM6 only display a promising degree of thermal stability in case of an active layer thickness relevant for lab-scale devices but not thicker active layers relevant for high-throughput coating techniques[21].

Here, we explore whether the use of acceptor mixtures that comprise more than two components can result in a substantial increase in thermal stability of the active layer. The use of acceptor mixtures with more than two components is motivated by our recent observation that blending of up to eight perylene derivatives can lead to mixtures with an unprecedented ability to form a molecular glass, driven by the formation of a high-entropy ordered liquid composed of perylene aggregates[22]. In the current work, up to five Y-series acceptors are mixed (see Fig. 1 for chemical structures), in analogy to bulk metallic glasses, which tend to comprise up to five elements[23–25]. The combination of several acceptors has a minimal effect on their electronic disorder and blending with the widely used donor polymer PM6 results in hexanary blends with best device efficiencies of 17.6%. The hexanary blends display a high degree of thermal stability, independent of the film thickness (up to 390 nm), resulting in an unaltered photovoltaic performance upon annealing at 130 °C for 23 days (552 h) in the dark and under inert conditions.

## Results

### Glass formation of non-fullerene acceptor mixtures

In a first set of experiments, we studied the thermal behavior of the donor polymer PM6 and the Y-series acceptors Y1, Y6, Y11, Y16 and Y18 (Fig. 1). These five acceptors feature a similar lowest-unoccupied molecular orbital (LUMO) energy ranging from 4.33 eV (Y1) to 4.50 eV (Y6; see Supplementary Table 1). Differential scanning calorimetry (DSC) first heating thermograms were recorded at a rate of 10 °C min[−1] for material solidified from chloroform. For the donor polymer PM6 we do not observe any transitions (Fig. 2a). Instead, the first heating thermograms of the acceptors Y1 and Y6 display a crystallization exotherm above 220 °C (for Y6 crystallization enthalpy $\Delta H_c = 8\,\mathrm{J\,g^{-1}}$), followed by a melting endotherm around 290 °C (for Y6 peak melting temperature $T_m = 290$ °C and melting enthalpy $\Delta H_m = 29\,\mathrm{J\,g^{-1}}$). We note that $\Delta H_c < \Delta H_m$, and thus we propose that as-cast Y1 and Y6 show a certain degree of order followed by further crystallization above 220 °C. The thermal behavior observed here is in agreement with previous reports, i.e., Y6 features a low degree of order when solidified from chloroform but crystallizes above its $T_g \approx 205$ °C (midpoint of the heat flow step reported for bulk Y6 in ref. 16 using a heating rate of 1000 K s[−1]) followed by melting just below 300 °C[16]. We note that an additional exotherm appears above $T_m$ in the here reported DSC thermograms (Fig. 2a), which we assign to degradation of the acceptor material, in agreement with thermal gravimetric analysis (TGA) thermograms reported by Xiao et al.[26]. The DSC thermograms of Y11, Y16 and Y18 only feature one distinct endothermic peak above 280 °C with $\Delta H_m = 30–42\,\mathrm{J\,g^{-1}}$ (Supplementary Table 2) indicating melting of an ordered phase that had formed during solidification from chloroform (Fig. 2a). We conclude that Y1 and Y6 are initially more disordered while the other acceptors crystallize upon solidification from solution.

A DSC first heating thermogram of a mixture of three acceptors, Y6:Y11:Y16 (Fig. 2a), reveals a broad endothermic peak at $T_m = 282$ °C with significantly lower enthalpy of fusion of $\Delta H_m = 14\,\mathrm{J\,g^{-1}}$, which we assign to melting of crystallites of Y6, Y11 and Y16. The addition of two more acceptors resulted in even lower values of $T_m = 256$ °C and $\Delta H_m = 10\,\mathrm{J\,g^{-1}}$ in case of the pentanary mixture Y1:Y6:Y11:Y16:Y18 (Supplementary Table 2). A similar trend is observed for two other pentanary acceptor mixtures (see Supplementary Fig. 2). The absence

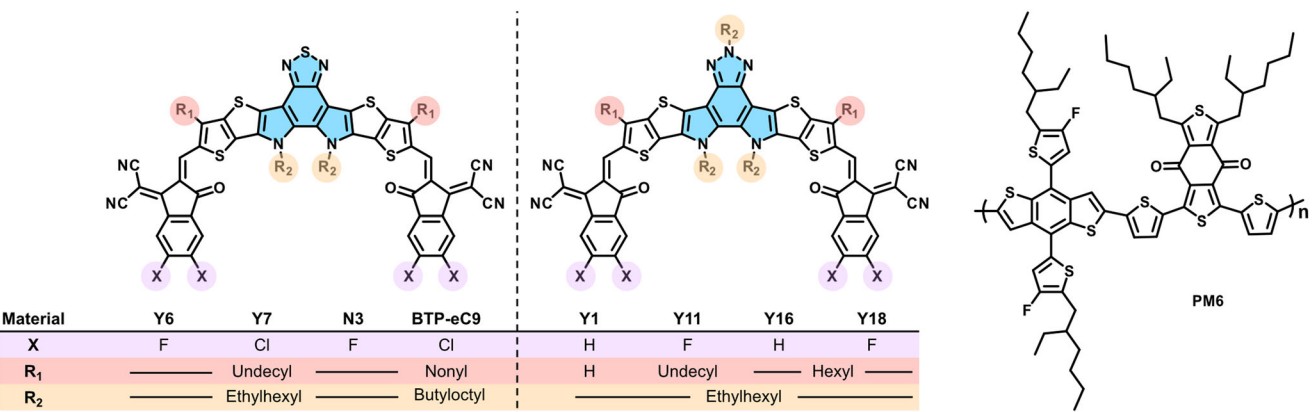

| Material | Y6 | Y7 | N3 | BTP-eC9 | Y1 | Y11 | Y16 | Y18 |
|---|---|---|---|---|---|---|---|---|
| X | F | Cl | F | Cl | H | F | H | F |
| R₁ | Undecyl | | | Nonyl | H | Undecyl | | Hexyl |
| R₂ | Ethylhexyl | | | Butyloctyl | | | Ethylhexyl | |

**Fig. 1 | Chemical structures of acceptor and donor materials.** Chemical structures of the Y-series acceptors and the donor polymer used for hexanary devices H1 and H5.

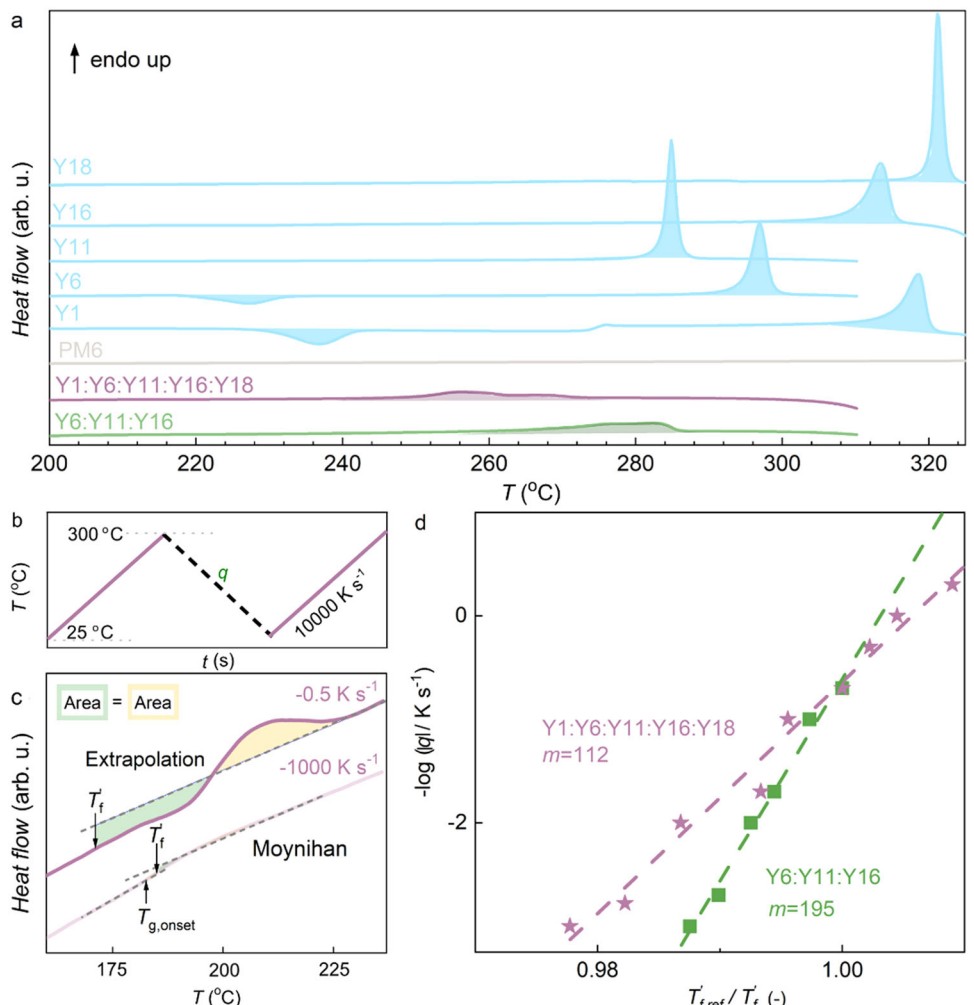

**Fig. 2 | Thermal analysis of acceptor mixtures. a** Differential scanning calorimetry (DSC) first heating thermograms of Y18, Y16, Y11, Y6, Y1, PM6, Y1:Y6:Y11:Y16:Y18 and Y6:Y11:Y16. **b** Fast scanning calorimetry (FSC) protocol. The material is first heated to 300 °C, then cooled to 25 °C with different cooling rates $q$ ranging from −0.5 to −1000 K s$^{-1}$ followed by heating with 10000 K s$^{-1}$. **c** Heating scans after fast (light lavender) and slow (dark lavender) cooling of Y1:Y6:Y11:Y16:Y18. **d** Fragility plot with −log|$q$| vs. $T'_{f,ref}/T'_f$ for Y1:Y6:Y11:Y16:Y18 (lavender stars) and Y6:Y11:Y16 (olive squares). The reference fictive temperature $T'_{f,ref}$ is taken from FSC thermograms recorded at 5 K s$^{-1}$. Source data are provided as a Source Data file.

of melting endotherms above 280 °C suggests that crystallization of single-acceptor crystallites is strongly suppressed (note that all Y6 polymorphs melt just below 300 °C[16]. We argue that mixing increases the entropy of the liquid state in case of ternary and pentanary mixtures, which reduces the driving force for crystallization (cf. introduction). The remaining broad endotherm with a lower $T_m$ and $\Delta H_m$ may arise because residual single-acceptor crystallites are still present, which are however smaller in size. Alternatively, aggregates of several acceptors may have formed, as discussed below.

Since the neat acceptors as well as the acceptor mixtures are able to crystallize to some extent upon solidification from solution, we wanted to find out under which conditions, if any, an entirely amorphous glassy state can be obtained. We chose to use fast scanning calorimetry (FSC) to cool thin films of the materials from 300 °C at vastly different rates ranging from $q$ = −0.5 to −1000 K s$^{-1}$ followed by rapid heating at 10000 K s$^{-1}$ (Fig. 2b). We were unable to record FSC thermograms for thin films of any of the single acceptors when cooled at up to $q$ = −1000 K s$^{-1}$ because the material lost contact with the sensor (not shown), likely because the material was able to crystallize. We note that Gutierrez-Fernandez et al. were able to record a FSC thermogram for a bulk Y6 sample after cooling at $q$ = −10000 K s$^{-1}$[27], which suggests that the critical cooling rate needed for reaching a

glassy state $q_{critical}$ is larger than −1000 K s$^{-1}$. We were able to record FSC thermograms for the Y6:Y11:Y16 ternary mixture (Supplementary Fig. 3). Material cooled at low rates of less than −5 K s$^{-1}$ had time to crystallize, as indicated by a broad melting peak around $T_m$ = 260 °C (cf. DSC thermogram in Fig. 2a and FSC heating scans in Supplementary Fig. 3). Instead, more rapidly cooled material formed glassy films suggesting a critical cooling rate $q_{critical}$ = −5 K s$^{-1}$ (Fig. 2b). Strikingly, the pentanary acceptor mixture Y1:Y6:Y11:Y16:Y18 formed a glassy film at any cooling rate between $q$ = −0.5 and −1000 K s$^{-1}$, indicating that $q_{critical}$ is less than −0.5 K s$^{-1}$ (Fig. 2c).

Since both the ternary and pentanary mixture readily form a glass when cooled faster than $q_{critical}$, the kinetic fragility index, $m$, could be determined according to:

$$m = \frac{-d \log|q|}{d(T'_{f,ref}/T'_f)}\bigg|_{T = T'_{f,ref}} \tag{1}$$

where $T'_f$ is the fictive temperature measured at cooling rate $q$ and $T'_{f,ref}$ is the fictive temperature measured at a low reference cooling rate, here obtained at −5 K s$^{-1}$ with FSC. The kinetic fragility describes how fast the relaxation dynamics of a material slow down during cooling in the proximity of its $T_g$ and can be used to classify a material

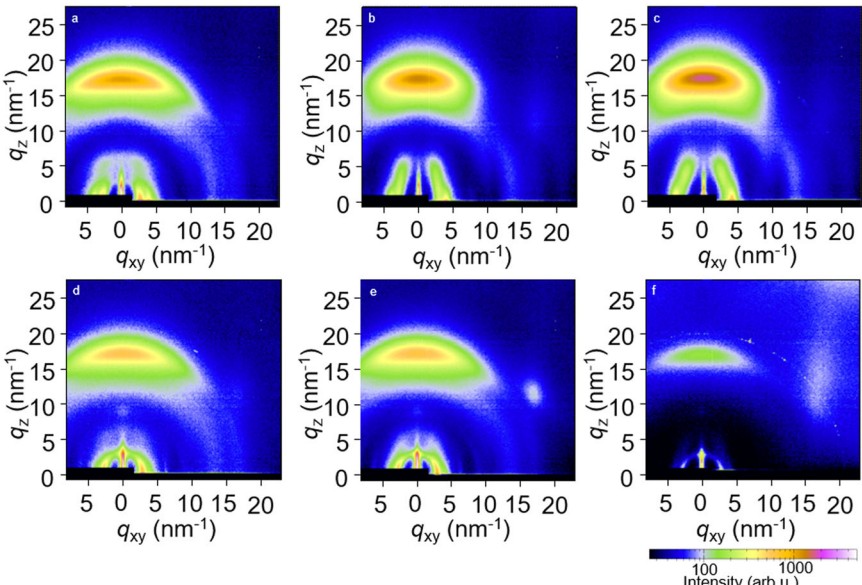

**Fig. 3 | Structural characterization of acceptor mixtures and bulk-heterojunction blends.** GIWAXS patterns of thin films of **a** Y6, **b** Y6:Y11:Y16, **c** Y1:Y6:Y11:Y16:Y18, **d** PM6:Y6, **e** PM6:Y6:Y11:Y16 and **f** PM6:Y1:Y6:Y11:Y16:Y18 spin-coated from chloroform and annealed at 120 °C for 1 min.

as a strong ($m < 40$) or fragile ($m > 90$) glass former. All regular materials relax over long periods of time to release excess energy, which can lead to gradual crystallization. Strong glass formers are characterized by a long relaxation time (high viscosity), which can suppress nucleation and growth of crystals even above $T_g$, and hence ease glass formation[27]. We constructed fragility plots for both mixtures and extracted fragility values of $m = 195$ and $112$ for the ternary and pentanary acceptor mixtures, respectively (Fig. 2d). Evidently, both mixtures are fragile and thus can be classified as poor glass formers, which can be expected to crystallize if given sufficient time, consistent with the melting endotherms observed with DSC (Fig. 2a).

Grazing-Incidence Wide-Angle X-ray Scattering (GIWAXS) allowed us to gain further insight into the nanostructure of the acceptor mixtures. GIWAXS patterns recorded for Y6 and the Y1:Y11:Y16 ternary mixture solidified from chloroform feature a strong in-plane diffraction at $q = 2.5$ nm$^{-1}$ indicating that the acceptor material adopts a partially ordered nanostructure, referred to as "as cast" by Gutierrez-Fernandez et al.[16]. For the Y1:Y6:Y11:Y16:Y18 pentanary mixture, instead, we observe a distinct in-plane diffraction at $q = 4.2$ nm$^{-1}$ as well as an off-axis diffraction at about 5 nm$^{-1}$ [28,29] (Fig. 3), which indicate that the acceptor material adopts the more ordered phase 1, again following the nomenclature proposed by Gutierrez-Fernandez et al.[16]. We argue that mixtures of Y-series acceptors form a more ordered phase composed of aggregates of randomly assembled derivatives, compared to neat Y6, which forms a more disordered phase (see Fig. 3a–c). This behavior is analogous to the formation of an ordered liquid by mixtures of up to eight structurally similar perylene derivatives, driven by an increase in entropy as a result of the many possible aggregate configurations[22]. We note that the here analyzed blend films can be considered as ordered glassy films (cf. glass formation inferred from FSC) with local structural order (cf. aggregates indicated by GIWAXS).

Blending of the donor polymer PM6 with Y6 or acceptor mixtures altered the GIWAXS diffraction patterns (Fig. 3d–f). GIWAXS patterns of the PM6:Y6 binary mixture (B1), the PM6:Y6:Y11:Y16 quaternary blend (Q1) and the PM6:Y1:Y6:Y11:Y16:Y18 hexanary blend (H1) feature distinct diffractions associated with PM6, e.g., at $q = 3$ nm$^{-1}$ (Fig. 3d–f). This observation indicates that the donor polymer has at least in part phase-separated into PM6-rich domains. Instead, the intensity of the in-plane diffraction at $q = 4$ nm$^{-1}$ is weak (Fig. 3d), suggesting that the addition of PM6 suppresses the formation of an ordered acceptor

phase to some extent (cf. ref. 16 for GIWAXS patterns of ordered phase 1). The Scherrer equation was used to estimate the coherence length of ordered domains, yielding a value of 11–14 nm for the PM6 domains and 9 nm for the acceptor blends, independent of the number of acceptors (Supplementary Table 3). We employed atomic force microscopy-based infrared spectroscopy (AFM-IR) to confirm the presence of polymer- and acceptor-rich domains in case of the hexanary blend H1. The FTIR spectra of PM6 show a unique peak at 1648 cm$^{-1}$ from alkene vibrations that we used to map the donor-rich domains in a thin film of the hexanary blend (Supplementary Fig. 4). We observe small donor-rich domains, likely surrounded by acceptor material, with a domain size of around 10 to 100 nm (Supplementary Fig. 4).

## Devices based on non-fullerene acceptor mixtures

OSCs based on binary, ternary, quaternary, pentanary and hexanary blends with a wide range of acceptor compositions (selected from Y1, Y6, Y11, Y16 and Y18 (see Fig. 1 for chemical structures) were fabricated with a conventional device architecture (see Methods section for details, $J$-$V$ curves of Q1, B1 and H1 based devices in Supplementary Fig. 5a and summary of device parameters in Table 1 and Supplementary Table 4). The ternary devices have a similar performance to the quaternary devices, whereas the average device performance of pentanary and hexanary devices is slightly lower (-5%) compared with the ternary devices. Also, Supplementary Fig. 6 depicts that hexanary devices with an equal weight ratio of acceptors (H1) feature a performance that is comparable to binary devices (e.g. B1, cf. Table 1) and on a par with other hexanary devices H2-4 that comprise different donor:acceptor ratios (cf. Supplementary Table 4). Therefore, we chose to compare devices comprising equal weight fractions of the acceptors in the remainder of this study.

Devices based on H1 and B1 blends display similar short-circuit current density $J_{sc}$ values (see Table 1 for details). The slightly higher open-circuit voltage $V_{oc} = 0.88$ V of H1 based devices compared to $V_{oc} = 0.85$ V in case of B1 based devices can be assigned to a larger driving force $\Delta($IE$_D$ - EA$_A)$, where IE$_D$ is the ionization energy of the donor polymer and EA$_A$ is the weighted average of the electron affinities of the acceptors[30,31]. We also note that H1 based devices feature a lower fill factor FF = 62.0% compared to B1 based devices with FF = 74.9%, which can be explained with an increase in energetic

disorder as a result of mixing of several acceptors with slightly different EAs[32].

To overcome this limitation, a second type of hexanary device, H5, was fabricated based on another set of five different Y-series acceptors (Y6, Y7, Y18, N3 and BTP-eC9; see Fig. 1 for chemical structures). We argue that the selection of acceptors with similar EAs, which range from 4.44 to 4.50 eV (Supplementary Table 1), leads to a lower degree of energetic disorder resulting in a higher FF = 71.2% for H5 devices, and thus an improved device performance with a PCE = 17.1% (Table 1 and Supplementary Fig. 5a). Doctor blading of the H5 active layer in air resulted in a PCE = 17.6%, which, considering the error in $J_{sc}$, is statistically similar to the performance of spin-coated devices (Table 1). The invariance of the device efficiency is consistent with AFM images of spin-coated and doctor-bladed films, which reveal a comparable surface topography (Supplementary Fig. 7). It can be concluded that OPV blends comprising a mixture of several acceptors can be processed with high-throughput printing techniques (Table 1 and Supplementary Fig. 5b).

Accelerated ageing experiments were carried out to assess the thermal stability of hexanary devices. Devices were annealed at 130 °C prior to spin-coating of a PNDIT-F3N electron transport layer and evaporation of the top Ag electrode (see "Methods" section for details). Both hexanary devices (H1, H5) retained their initial photovoltaic performance even after annealing of the active layer for 23 days at 130 °C (see Fig. 4a, b and Supplementary Fig. 8). In contrast, the PCE of binary devices B1 and B2 decreased by about one-third compared to their initial PCE after 23 days of ageing (see Fig. 4a, b and Supplementary Fig. 8). The decrease in the PCE of binary devices is largely due to a considerable drop in FF by one-fifth (Fig. 4a).

To understand if the use of multi-component mixtures can be extended to other types of acceptors, similar annealing studies were conducted for hexanary devices based on ITIC derivatives (H6; for device parameters see Supplementary Table 5 and Supplementary Fig. 9). In contrast to hexanary devices based on Y6-type acceptors, the initial efficiency of H6 devices is one-third lower than corresponding binary devices (B6-B8; Supplementary Table 5). The lesser complementary absorption among the ITIC derivatives and the donor polymer PM6 (Supplementary Fig. 10) is the likely cause for the lower $J_{sc}$ of H6 devices compared with the binary devices B6-B8 (Supplementary Table 5). Further, the ITIC derivatives that make up H6 devices feature an up to 0.41 eV difference in $EA_A$ (see Supplementary Table 1), which results in a relatively high degree of energetic disorder that limits the device performance. Gratifyingly, H6 devices maintain their performance when annealing the active layer for 23 days at 130 °C, while corresponding binary devices display a significant decrease in device efficiency (Supplementary Fig. 11). As a result, after only 5 days of annealing at 130 °C the efficiency of H6 devices exceeds that of, e.g., B8 devices. Evidently, mixing of a large number of acceptors is a strategy that can be extended to other classes of NFAs (Supplementary Fig. 11).

We carried out time delayed collection field (TDCF) experiments to understand how ageing of the active layer influences charge

## Table 1 | Device parameters

| Device active layer | $J_{sc}$ (mA cm$^{-2}$) | $V_{oc}$ (mV) | FF (%) | PCE (%) |
|---|---|---|---|---|
| B1 | 26.2 ± 0.5 | 850 ± 20 | 74.9 ± 2 | 16.7 ± 0.5 |
| T1 | 22.0 ± 0.4 | 899 ± 2 | 64.0 ± 1 | 12.7 ± 0.2 |
| Q1 | 23.0 ± 0.6 | 880 ± 10 | 58.7 ± 3 | 11.9 ± 0.9 |
| P1 | 22.0 ± 0.1 | 902 ± 2 | 60.3 ± 0.5 | 11.98 ± 0.1 |
| H1 | 25.4 ± 0.3 | 880 ± 5 | 62.0 ± 1 | 13.7 ± 0.4 |
| H5 | 27.3 ± 0.5 | 880 ± 1 | 71.2 ± 1 | 17.1 ± 0.4 |
| H5[a] | 28.3 ± 0.5 | 870 ± 2 | 73.0 ± 1 | 17.6 ± 0.4 |

Mean and standard deviation of photovoltaic device parameters for 6 as-cast devices on the same substrate with the following active layer compositions: Binary (B1): PM6:Y6 (1:1.2); Ternary (T1): PM6:Y1:Y18 (1:0.6:0.6); Quaternary (Q1): PM6:Y6:Y11:Y16 (1:0.4:0.4:0.4); Pentanary (P1): PM6:Y1:Y6:Y11:Y16 (1:0.3:0.3:0.3:0.3); Hexanary-1 (H1): PM6:Y1:Y6:Y11:Y16:Y18 (1:0.24:0.24:0.24:0.24:0.24) and Hexanary-5 (H5): PM6:Y6:Y7:Y18:N3:BTP-eC9 (1:0.24:0.24:0.24:0.24:0.24). The donor:acceptor ratios are given as weight ratios.
[a]Fabricated by doctor blading in air. The active area of each pixel is 0.1 cm$^2$.

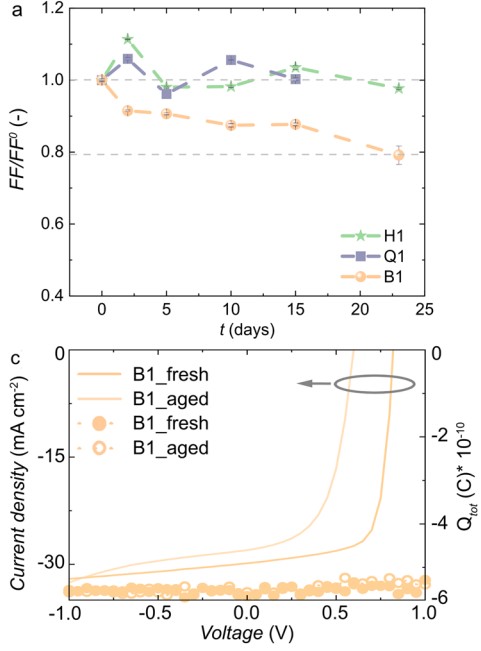

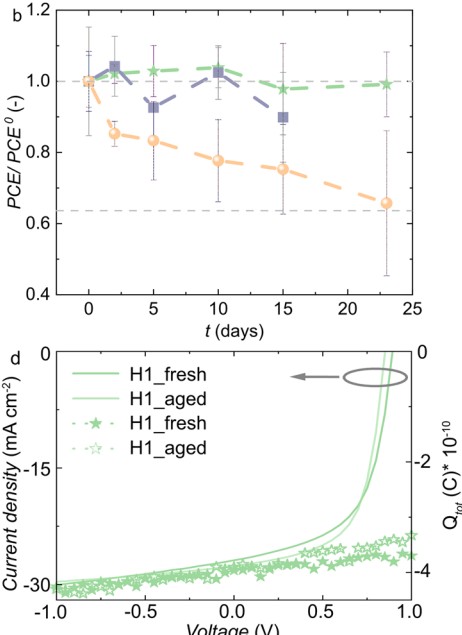

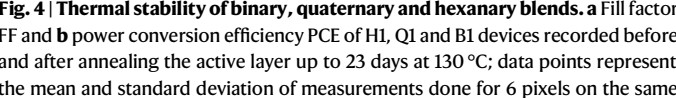

**Fig. 4 | Thermal stability of binary, quaternary and hexanary blends. a** Fill factor FF and **b** power conversion efficiency PCE of H1, Q1 and B1 devices recorded before and after annealing the active layer up to 23 days at 130 °C; data points represent the mean and standard deviation of measurements done for 6 pixels on the same substrate. The active area of each pixel is 0.1 cm$^2$. Time delayed collection field measurements with the total extracted charge $Q_{tot}$ (doted lines) as a function of the applied pre-bias overlaid with the respective J-V curves (solid lines) for **c** the fresh and aged B1 and **d** H1 devices. Source data are provided as a Source Data file.

separation and recombination processes and thus the device parameters. Charge generation in case of H1, H5 and B1 devices displays a slight field dependence but is not affected by ageing of the respective devices (Supplementary Fig. 12) and thus cannot explain the observed loss in FF, which is most pronounced in case of binary devices (see Supplementary Fig. 8). However, devices based on aged B2 blends showed a clear change in the bias dependence of the charge generation, which is consistent with the loss in $J_{sc}$ observed upon annealing (Supplementary Fig. 8a). Interestingly, both binary (B1, B2) and hexanary (H1, H5) devices feature a similar recombination constant prior to and after ageing (Fig. 4c, d). The similar charge generation and extraction properties of as-cast and aged devices suggest that the loss in FF of B1 based devices is due to a higher series resistance in the aged active layer compared with the respective as-cast B1 active layer[33] (Supplementary Fig. 13). In addition, Wöpke et al. have assigned the increase in trap density upon annealing to the crystallization of PM6[34].

GIWAXS of as-cast and annealed active layers was carried out to gain further insight into structural changes, especially the size of ordered polymer and acceptor domains. GIWAXS patterns of B1 as well as Q1 and H1 blend films, annealed at 130 °C for 120 h (Supplementary Fig. 14), indicate that the coherence length of ordered polymer domains increases for all samples. PM6 orders independent of the number of acceptors, likely within the PM6-rich domains inferred from AFM-IR (see Supplementary Fig. 4), and therefore changes in the trap density due to crystallization of the polymer cannot explain the loss in FF in case of binary devices.

The acceptor domains of B1 blend films do not change upon annealing, as indicated by a similar coherence length of ordered domains of about 9 nm (see Supplementary Table 3). In contrast, GIWAXS patterns of annealed Q1 and H1 blend films reveal a strong increase in the diffraction at $q = 4$ nm$^{-1}$ (indicative of ordered phase 1 discussed in ref. 16) accompanied by an increase in coherence length from 9 nm to 27 and 16 nm, respectively (Supplementary Table 3). Fullerene derivatives[35] and ITIC derivatives[15] are able to undergo local spatial rearrangements far below their $T_g$ during prolonged thermal annealing of BHJ blends. Likewise, Y-series acceptors appear to experience aggregation, which however is more pronounced in case of acceptor mixtures. It is feasible that the various acceptor molecules assemble into clusters similar to perylene mixtures that aggregate to form a high-entropy liquid[22]. We argue that (the degree of change in) the size of ordered acceptor domains is not a good metric for judging the performance and thermal stability of devices that comprise mixtures of Y-series acceptors. Instead, changes that are not captured by GIWAXS likely cause the decrease in FF of B1 devices upon thermal annealing, such as changes in the composition of the disordered phase(s). As discussed below, an alternative explanation for the drop in FF in PM6:Y6 devices (B1) is the purification of PM6-rich domains upon annealing for long durations (Y6 has a low miscibility of only 8.5% in PM6), as proposed by Qin et al.[17].

### Impact of active layer thickness on thermal stability

In a further set of experiments, we examined the impact of the active-layer thickness on the thermal stability of H1 and B2 devices with three different active layer thicknesses each (see Supplementary Table 7 for thickness values). Among these 6 different devices, H1_3 and B2_3 devices were doctor-bladed in air, and the other four types of devices were spin-coated in a glovebox (see Supplementary Table 8 for device performances). Both, binary and hexanary devices with thicker films (H1_3 and B2_3) suffer from a lower $V_{oc}$ compared to the respective thinner active layer film devices. This is likely due to the higher charge recombination in the thicker active layer films owing to the limited exciton diffusion lengths of the NFAs[36,37]. In addition, B2_3 devices suffer from a lower $J_{sc}$ compared with the thinner binary devices (B2_1 and B2_2), whereas H1_2 and H1_3 devices feature similar $J_{sc}$ values.

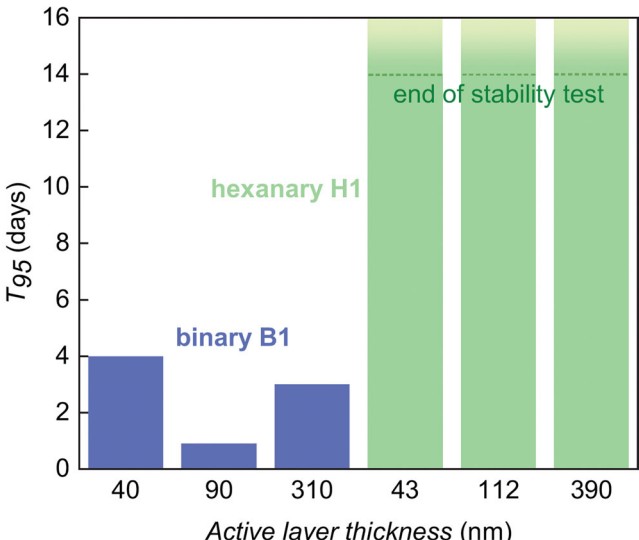

**Fig. 5 | Impact of active layer thickness on thermal stability.** The time $T_{95}$ for binary (B2) and hexanary devices (H1) to reach 95% of their initial PCE upon annealing at 130 °C, measured for different active layer thicknesses (for H1 devices $T_{95}$ had not been reached at the end of the 14-day stability test). The active area of each pixel is 0.1 cm². Source data are provided as a Source Data file.

All devices were annealed at 130 °C for 14 days in the dark and under an inert atmosphere (see Methods section for details). The thermal stability of B2 devices showed a significant dependence on the active layer thickness (Fig. 5; more details can be found in Supplementary Table 8 and Supplementary Figs. 15–16). While B2 devices with thin active layers (B2_1) retain their photovoltaic performance upon annealing, in case of thick B2 (B2_2) devices we observe an 86 % drop in PCE upon annealing at 130 °C for 14 days, predominantly due to a loss in $J_{sc}$ and FF. The poor $J_{sc}$ and FF of the as-casted B2_3 devices likely masked a further drop in photovoltaic performance upon annealing, which is reflected in the slightly longer $T_{95}$. The observed thickness dependency is in agreement with the recent work by Min et al. who have suggested that the thickness-dependent thermal stability of binary and ternary devices may occur due to a $T_g$ that changes with the active layer thickness[21]. The latter type of behavior has been observed experimentally for ternary blend films composed of a thiophene-quinoxaline polymer and two fullerene acceptors that showed a gradual increase in $T_g$ with decreasing film thickness, which was most pronounced for thin films with a thickness of less than 150 nm[38].

Strikingly, devices comprising the hexanary blend H1 showed a thickness-independent thermal stability, i.e., devices with an active layer annealed at 130 °C for 14 days featured a similar PCE as as-cast devices. Evidently, the superior thermal ability of hexanary blends also persists in the case of 390 nm thick films. This observation suggests that the here proposed strategy, i.e., mixing of a multitude of acceptors, may facilitate the fabrication of thermally stable devices via high-throughput processing techniques, which tend to yield thick active layers.

## Discussion

We have demonstrated that photovoltaic devices fabricated with five-component acceptor mixtures and a donor polymer feature significantly improved thermal stability upon prolonged annealing of the BHJ active layer at a high temperature of 130 °C, which is vital for upscaling and a longer device lifetime. The proposed strategy of mixing a large number NFAs is generic as evidenced by the improved stability of hexanary blends based on both Y- and ITIC-type acceptors.

While pentanary mixtures of Y-series acceptors form a glassy phase they display a very high kinetic fragility of $m = 112$, and thus can be characterized as fragile glass formers. As a result, thermal annealing leads to the formation of small, ordered acceptor domains, which can be expected to benefit charge extraction since the ordered phase 1 of Y6 has a slightly higher electron mobility (measured with field-effect transistors) as compared to its more disordered as-cast phase[16]. The active layer of hexanary devices is more resilient to thermal annealing compared to binary devices, which tend to suffer from a loss in charge generation efficiency and/or an increase in series resistance (Supplementary Fig. 10) upon thermal annealing. Qin et al. have argued that Y6:PM6 binary devices degrade upon annealing as a result of purification of polymer-rich mixed domains due to the low miscibility of Y6 in PM6[17]. It is feasible that acceptor mixtures yield a slightly higher overall miscibility in the polymer-rich phase, in analogy to the higher miscibility of, e.g., $C_{60}$:$C_{70}$[39] or perylene diimide mixtures[9] in organic solvents, which would positively impact device stability.

Crucially for high-throughput coating, superior stability is also observed upon thermal annealing of thicker active layers, which is difficult to achieve with binary or ternary blends[21]. To achieve a high FF, and hence PCE with multi-component acceptor mixtures, it is important to select acceptors that are not only structurally similar but also feature a similar EA, which minimizes energetic disorder. As a result, doctor-bladed hexanary devices based on mixtures of judiciously selected NFAs feature a PCE of 17.6%. In summary, BHJ blends that comprise a considerably larger number of acceptors than what has been studied previously, i.e., more than two or three components, can offer not only a state-of-the-art device efficiency but also superior thermal stability, paving the way for stable organic photovoltaics.

In conclusion, we have investigated the phase behavior of 3- and 5-component acceptor mixtures based on Y-type acceptors. Thermal analysis and GIWAXS revealed that acceptor mixtures form a glassy phase comprising small aggregates. The structural disorder is not detrimental to the performance of photovoltaic devices, provided that acceptors with similar EAs are mixed, which reduces the degree of energetic disorder. Consequently, hexanary devices with judiciously selected components displayed a power conversion efficiency of up to 17.6%. Further, hexanary blends were able to retain their photovoltaic performance upon annealing for at least 23 days at 130 °C in the dark and an inert atmosphere, irrespective of the active layer thickness. The here proposed strategy of mixing a large number of acceptors can result in a homogeneous BHJ blend nanostructure with a thickness-invariant thermal stability, which is urgently needed for upscaling and ultimately commercialization of organic photovoltaics. While the impact of mixing on the thermal stability is applicable to different types of NFA mixtures (e.g. Y- and ITIC-type acceptors), the optimal number of acceptors in a mixture may vary for different types of NFAs, thus providing an additional parameter that can be adjusted, leading to further gains in stability and device performance. Given the vast number of possible acceptor combinations, evaluating a wider range of multi-component devices becomes prohibitively labor intensive. It can be anticipated that this work inspires follow-up studies that use high-throughput robot-based device fabrication[40] or the characterization of devices with a composition gradient[41] to identify blend compositions that offer both optimal device performance and stability among multi-component devices.

## Methods

### Materials
The donor polymer, PM6 (PBDBT-2F; number-average molecular weight $M_n \approx 39$ kg mol$^{-1}$, polydispersity index PDI = 2.4), the acceptor materials Y1, Y6, Y7, Y11, Y16, Y18, N3, BTP-eC9, ITIC, ITIC-Th, ITIC-M, ITCC, ITIC-4F and ITIC-4Cl as well as the electron transport layer material PNDIT-F3N were purchased from Solarmer and used as received. Poly(3,4-ethylenedioxythiophene) polystyrene sulfonate, PEDOT:PSS

(P VP.AI 4083), dispersion was purchased from Heraeus. Chloroform (purity > 99%), acetone (purity > 99%), acetic acid (purity > 99.9%), chloronaphthalene (purity > 85%) and anhydrous isopropyl alcohol (IPA; purity > 99.5%) were obtained from Sigma Aldrich and used as received. All samples were processed from 12 g L$^{-1}$ chloroform solutions. For more information, refer to device fabrication and characterization.

### Differential scanning calorimetry (DSC)
DSC measurements were carried out with a Mettler Toledo DSC2 equipped with a gas controller GC 200. Around 4 mg of material was collected in 40 µL DSC Al crucibles. Two heating and cooling cycles between 25 and 350 °C were carried out at a rate of 10 °C min$^{-1}$.

### Fast scanning calorimetry (FSC)
A Mettler Toledo Flash DSC 1 was used for the fragility measurements. The solutions were drop cast on the FSC chip sensors and dried at ambient conditions. Samples were first heated to 300 °C and then cooled to 0 °C with cooling rates ranging from −0.1 to −1000 K s$^{-1}$. Finally, samples were heated with 10,000 K s$^{-1}$ (see FSC protocol in Fig. 2b). To calculate the limiting fictive temperature $T_f'$, Moynihan's matching area method (Richardson's method in the Mettler Toledo software) was used if $T_f'$ was above the onset of $T_g$, (Fig. 2c):

$$\int_{T_f'}^{T \gg T_g} \left( C_{pl} - C_{pg} \right) dT = \int_{T \ll T_g}^{T \gg T_g} \left( C_p - C_{pg} \right) dT$$

where $C_{pl}$ is the heat capacity of the liquid, $C_{pg}$ is the heat capacity of the glass and $C_p$ is the apparent heat capacity of the sample. If the $T_f'$ was below the onset of $T_g$, a simplified extrapolation equation was used: $\int_{T_f'}^{T \gg T_g} \left( C_{pl} - C_{pg} \right) dT = 0$ (Fig. 2c).

### Grazing-incidence wide-angle X-ray scattering (GIWAXS)
GIWAXS diffractograms were collected at the CMS beamline at NSLS II, Brookhaven National lab. An X-ray beam (13.5 keV) was guided onto the sample substrate at an incidence angle of 0.15°. A Pilatus 800k detector was used to collect the diffractograms at an exposure period of 10 s.

### Atomic force microscopy-based infrared spectroscopy (AFM-IR)
AFM-IR measurements were conducted with a nanoIR3 instrument (Bruker) equipped with an MIR-cat QT 2400 QCL infrared laser from Daylight Solutions. Samples were prepared by spin coating on cleaned silicon wafers and gold coated silicon tips were used for scanning the samples.

### Atomic force microscopy (AFM)
AFM measurements were conducted on the respective blend films using MD-NDT with OTESPA cantilevers from Bruker (nominal tip radius 10 nm) and the AFM images (5 µm × 5 µm scan) were on recorded in semi-contact mode. All the films were prepared by following the same conditions as the device fabrication.

### PESA
Photoelectron spectroscopy in air (PESA) measurements were recorded using a Riken Keiki PESA spectrometer (Model AC-2) with a power setting maximum of 50 nW. The films for PESA were spin-coated on glass substrates.

### Device fabrication and characterization
12 g L$^{-1}$ each of the donor (PM6) and acceptor materials were dissolved in chloroform separately for a maximum of 2.5 h at 40 °C. The appropriate ratios of donor and acceptor solutions were mixed in a new vial at the same temperature. The active layer solution was completed by addition of 0.5% (V/V) of chloronaphthelene to the blend

solution. The electron transport layer solution was prepared by stirring 1.5 g L$^{-1}$ of PNDIT-F3N in a solvent mixture of IPA and 0.5% of acetic acid at room temperature for one hour. The hole transport layer solution D-PEDOT was prepared by stirring one volume part of PEDOT:PSS dispersion with three volume parts of IPA at room temperature for a minimum of 10 min.

Pre-patterned Indium Tin Oxide (ITO) glass substrates were cleaned in an ultrasonic bath with acetone and IPA, and dried under nitrogen flow. The D-PEDOT layer was spin-coated at 4000 rpm for 40 s, followed by annealing for 1 min at 120 °C in air on the patterned and cleaned ITO. Active layers were spin-coated at 1750 rpm inside a glove box, followed by annealing at 120 °C for 1 min. Finally, a PNDIT-F3N layer was spin-coated at 1250 rpm for 30 s followed by evaporation of a 100 nm thick layer of silver at $1 \times 10^{-6}$ bar. $J$-$V$ curves were recorded using a Keithley 2400 source meter and a WaveLabs sinus-70 solar simulator calibrated to 1 sun, AM1.5 G.

### Thickness measurements

Thickness measurements were conducted using a mechanical profiler, KLA Tencor P-6. D-PEDOT was spin-coated and annealed on a glass slide with the same fabrication parameters as described above. Later, the active layer was spin-coated on top of D-PEDOT as described above. The mechanical probe scanned across the scratched area to measure the thickness of the stack. The active layer thickness was calculated by subtracting the D-PEDOT layer thickness from the total thickness.

### Time-delayed collection field (TDCF) measurements

A home-built TDCF setup was used to reveal potential field dependence on charge generation in solar cells. The optical excitation used the second harmonic (532 nm) of an actively Q-switched sub-ns Nd:YVO$_4$ laser (INNOLAS piccolo AOT) operating at 5 kHz as excitation with a pulse length of 1 ns. To minimize the RC response time (typically around 2 ns), a small device area of 1 mm$^2$ was used. The devices were measured under dynamic vacuum conditions to avoid any photo-degradation. Furthermore, the devices were held under pre-bias ranging from −4 V to the $V_{oc}$ during the photo-excitation pulse. The measurements were executed at low excitation fluences (0.1 μJ cm$^{-2}$) to reduce the impact of nongeminate recombination prior to charge collection. A collection voltage of −4 V was applied 10 ns after the laser pulse, prior to the expected onset of non-geminate recombination at low fluences. A Keysight S1160A functional generator was used to provide the pre-bias $V_{pre}$ and extraction bias $V_{coll}$, while a Keysight (MSOX3034T, 350 MHz) four channel digital oscilloscope was used to measure the current response of the solar cell.

### Thermal stability measurements

All thermal stability tests were conducted inside a glove box. The partial device stack (ITO/D-PEDOT/active layer) was annealed at the aforementioned temperatures. The device fabrication was completed by spin-coating PNDIT-F3N inside a glove box and evaporating silver as mentioned above. The devices were measured under a WaveLabs sinus-70 solar simulator calibrated to 1 sun, AM1.5 G.

### Reporting summary

Further information on research design is available in the Nature Portfolio Reporting Summary linked to this article.

## Data availability

The data that support the findings of this study are available from the corresponding author upon request. The source data used to prepare Figs. 2, 4 and 5 as well as Supplementary Figs. 2–13, 15 and 16 are provided with this paper. Source data are provided with this paper.

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

## Acknowledgements

This publication is based upon work supported by the King Abdullah University of Science and Technology (KAUST) Office of Sponsored Research (OSR) under Award No: OSR-CARF/CCF-3079 and OSR-CRG2018-3746. S.H., E.J., C.M., I.J. and E.M. gratefully acknowledge financial support from the Knut and Alice Wallenberg Foundation through the project "Mastering Morphology for Solution-borne Electronics" (grant number 2016.0059). S.H. and C.M. acknowledge support from the King Abdullah University of Science and Technology (KAUST) Office of Sponsored Research (OSR) under grant agreement no. OSR-2019-CPF-4106. The authors thank the National Synchrotron Light Source II (NSLS-II, Contract No. DE-SC0012704), Brookhaven National Laboratory for providing GIWAXS experiment time. S.H.K.P. acknowledges the timely help of L. Lanzetta with hyperspectral PL imaging studies.

## Author contributions

S.H.K.P. fabricated devices, performed thermal stability studies, analyzed data. S.H.K.P. prepared the first draft of the manuscript with contributions from S.H., Y.W., J.H., H.X., D.R.V. and A.S. assisted with thermal stability studies. S.H. performed the FSC and DSC studies and analysis. E.J. prepared GIWAXS samples and assisted with FSC studies. I.J. and S.H. performed AFM-IR and analysis, supervised by E.M. S.C. conducted the TDCF studies, supervised by F.L. J.K and J.G analyzed the TDCF measurements. D.R.V. performed AFM measurements. S.H.K.P. and Y.W. conducted the thickness measurements. R.L. performed the GIWAXS measurements and L.Y. analyzed the GIWAXS data. C.M. conceived the study, supervised the thermal analysis, and revised the manuscript. D.B. supervised and directed the device-based studies and revised the manuscript. All authors contributed to the revision of the final version of the manuscript.

## Funding

## Competing interests

The authors declare no competing interests.
