## [Peer Review File · Nature Communications]

Hexanary Blends: A Strategy Towards Thermally Stable Organic PhotovoltaicsREVIEWER COMMENTS

Reviewer #1 (Remarks to the Author):

In the manuscript 'Hexanary Blends: A Strategy Towards Thermally Stable Organic Photovoltaics', the authors reported a generic strategy for improving the thermal stability of non-fullerene-based bulk heterojunction organic solar cells. By blending five Y-series acceptors, which feature a similar chemical structure and comparable electron affinities with a donor material, the resulting hexanary bulk heterojunction organic solar cell achieved a high power conversion efficiency of up to 17.6%; more importantly, the performance of the hexanary device can remain unaffected under thermal annealing for 23 days at 130 °C in the dark and an inert atmosphere.

Overall, the experimental study, including material characterization, device fabrication, and performance test was implemented systematically, the results supported their conclusion. But the following issues need to be addressed before the manuscript can be accepted.

1. Five Y-series acceptor materials have been studied, in the binary (Y6: PM6), the quaternary (Y6:Y11:Y16:PM6), and the hexanary (Y1:Y6:Y11:Y16:Y18: PM6) solar cells. Why the authors only chose the combination of Y6:Y11:Y16 for the quaternary device? Did the other three-acceptor combinations have been tested, and did they demonstrate a similar performance?
2. Following up on the above question, did the authors test ternary (i.e., two acceptors + donor) devices and pentanary (i.e., four acceptors + donor) devices?
3. All acceptors were mixed with equal weight ratios in this study. Did the authors evaluate other weight ratios? Or say, whether this generic strategy depends on the weight ratio in the acceptor mixture as well?
4. On page 13, the authors mentioned that this multiple-component strategy has been proved with another type of acceptor, ITIC derivatives, where a similar thermal stability improvement has been observed. But the device performances summarized in Table S4 also showed that in the ITIC system, the performance of hexanary-3 device was poorer than all binary devices. Is this efficiency loss a potential issue that should be taken into consideration when using the multiple-acceptors strategy?

5. Devices made by doctor-blade casting in ambient conditions achieved even higher power conversion efficiency. Did the authors have any explanation for the better performance of the doctor-blade casting device?

Reviewer #2 (Remarks to the Author):

Authors demonstrate that hexanary blends (5 acceptors + 1 donor) improve the thermal stability of organic PV cells. Indeed, one major cause of degradation is crystallization of the small molecules NFA, which the increase of entropy provided by blending multiple acceptors together forbids. The authors' claim is witnessed in detail by several techniques like TDCF, GIWAXS and FSC.

The manuscript is well written and convincing, and the report is original and sound.

A few details should be considered to improve the manuscript:

- what was the area of the devices under test?

- we understand that increasing the entropy of mixing of a mixture favors the liquid (or glassy state - which is a frozen liquid). However, authors claim that the composition under study has "unprecedented ability to form a molecular glass". It is clear that the hexanary blend was unprecedented, but revealing the interest of glass formers to improve the stability of PSCs was not unprecedented, as reported in the review 'Revisiting the Optimal Nano-Morphology: Towards Amorphous Organic Photovoltaics' (DOI: 10.1002/tcr.201800158).

Reviewer #3 (Remarks to the Author):

In this manuscript, D. Baran and co-authors constructed a hexanary organic photovoltaic and conducted a systematic study on the thermal stability of the device. The authors demonstrated that the hexanary blend has better thermal stability, and tried to explain the mechanism behind it. This manuscript is data-rich but not innovative. However, if it wants to be published in Nature Communications, the reviewer consider at least that the following issues need to be discussed:

Page 6, line 139, the authors consider that the melting enthalpy of the material can be equivalent to the total amount of crystallization. What are the prerequisites for comparing different acceptor films?

Page 12, The authors claim that blade coating can achieve better device performance than spin coating, which indicates that the processing method has impacts on the morphology and performance of devices. However, when discussing the film thickness-dependent stability, the authors ignore the

morphology difference caused by different spin-coating speeds and solution concentrations, and the resulting impact on stability. Please add this discussion.

Page 13, For binary devices, the total extracted charge shows the same trend with bias voltage in fresh and aged devices, and the same phenomenon was also observed in hexanary devices (Fig. 4c and Fig. 4d). But what confuses the reviewers is the author's claim that "the loss in FF of B1 based devices is due to a decrease in the charge transport properties obtained upon ageing of the active layer", Please explain in detail. In addition, Fig. 4 and Fig. S9 suggest that H1, H2 exhibit bias-dependent charge extraction, why the authors claim that "Devices based on H1, H2, and B1 blends do not exhibit any changes in the bias dependence of charge generation" in page 14?

The thermal aged film (Q, H1) shows larger CL values of acceptor and PM6 than fresh film, however, the aged and fresh B1 film shows similar CL values of acceptor and PM6 (Table S3). This seems to indicate that B1 has better thermal stability, so why would the device based on B1 film be less thermally stable?

Page 16, The authors claim that "the superior thermal ability of hexanary blends also persists in the case of 117 nm thick films... may facilitate the fabrication of thermally stable devices via high-throughput processing techniques." However, the communities generally consider that 300nm is the minimum thickness requirement for industrial production, hence it is recommended to supplement the stability of 300nm devices or modify the language.

Page 11, In order to visually compare 2D GIWAXS images, the intensity scale should be added.

Page 14, line 279, the coherence length summarized in Table 1 or Table S3?

juni 27, 2023

Dear reviewer,

We would like to express our gratitude for carefully reviewing our manuscript. We took the time to address all comments in detail and, as a result, were able to significantly improve our manuscript. A description of all new measurements and modifications to our manuscript is provided in our point-by-point response (see below). Following the suggestions by the referees, performance data of 504 pixels (one pixel is equivalent to one solar cell) has been added to ratify our claims. The newly recorded data includes thirteen different active layer blends like, ternary, quaternary, pentanary and hexanary blends with varied acceptor combinations and weight ratios. Further, thermal stability data for 24 pixels of hexanary and binary active layer blends with thicknesses above 300 nm is provided. Lastly, thickness measurements of the active layers as well as absorbance spectra and atomic force microscopy images of hexanary and binary blends are added to elucidate the reported devices. We are looking forward to your feedback.

Yours sincerely,

Christian Müller & Derya Baran

Reviewer #1 (Remarks to the Author):

In the manuscript 'Hexanary Blends: A Strategy Towards Thermally Stable Organic Photovoltaics', the authors reported a generic strategy for improving the thermal stability of non-fullerene-based bulk heterojunction organic solar cells. By blending five Y-series acceptors, which feature a similar chemical structure and comparable electron affinities with a donor material, the resulting hexanary bulk heterojunction organic solar cell achieved a high power conversion efficiency of up to 17.6%; more importantly, the performance of the hexanary device can remain unaffected under thermal annealing for 23 days at 130 °C in the dark and an inert atmosphere.

Overall, the experimental study, including material characterization, device fabrication, and performance test was implemented systematically, the results supported their conclusion. But the following issues need to be addressed before the manuscript can be accepted.

1. Five Y-series acceptor materials have been studied, in the binary (Y6: PM6), the quaternary (Y6:Y11:Y16:PM6), and the hexanary (Y1:Y6:Y11:Y16:Y18: PM6) solar cells. Why the authors only chose the combination of Y6:Y11:Y16 for the quaternary device? Did the other three-acceptor combinations have been tested, and did they demonstrate a similar performance?

We thank the reviewer for the comment. Two new quaternary systems (PM6:Y6:Y11:Y18 (1:0.4:0.4:0.4), PM6:Y1:Y16:Y18 (1:0.4:0.4:0.4)) were measured; data were added to the SI, in Fig. S6 and Table S4. We have added the following paragraph in the main text:

"The ternaries have similar performances compared to the quaternaries, whereas the average device performance of pentanaries and hexanaries have slightly lower (approx. 5%) performance than the ternary devices."

2. Following up on the above question, did the authors test ternary (i.e., two acceptors + donor) devices and pentanary (i.e., four acceptors + donor) devices?

We have measured additional ternary, pentanary, and hexanary devices and summarize the device parameters in the SI, in Fig. S6 and Table S4. We have added the following paragraph to the main text to summarize the results:

"The ternary devices have a similar performance to the quaternary devices, whereas the average device performance of pentanary and hexanary devices is slightly lower (approx. 5%) compared with the ternary devices. Also, Fig. S6 depicts that hexanary devices with an equal weight ratio of acceptors (H1) feature a performance that is comparable to binary devices (e.g. B1, cf. Table 1) and on a par with other hexanary devices that comprise different donor:acceptor ratios (cf. Table S4). Therefore, we chose to compare devices comprising equal weight fractions of the acceptors in the remainder of this study."

Fig. S1. Composition-device efficiency diagram of binary (circles), ternary (triangles), quaternary (rhombuses), pentanary (pentagons) and hexanary (hexagons) devices based on PM6 and different combination of the acceptors Y6, Y1+Y18 and Y11+Y16. In brackets: # of compositions. The composition in blue color represent equal acceptor weight ratio.

Table S1. **Photovoltaic device parameters:** H1 based combinations. Mean and standard deviation of photovoltaic device parameters for 6 devices on the same substrate. B1-B5, T1-T2, Q1-Q3, P1-P2, H1-H4. The donor: acceptor ratios are weight ratios.

Device name	Donor: acceptor ratios	J_{sc} (mA cm ⁻²)	V_{oc} (mV)	FF (%)	PCE (%)
B1	PM6:Y6	26.2 ± 0.5	850 ± 20	74.9 ± 0.02	16.7 ± 0.5
B2	PM6:Y18	25.7 ± 0.1	880 ± 5	68.1 ± 0.2	15.4 ± 0.2
B3	PM6:Y1	12.6 ± 0.4	900 ± 7	51.5 ± 0.8	5.9 ± 0.2
B4	PM6:Y11	24.9 ± 0.5	860 ± 2	62.0 ± 0.5	13.3 ± 0.3
B5	PM6:Y16	8.6 ± 0.1	980 ± 4	46.7 ± 0.5	3.9 ± 0.1
T1	PM6:Y1:Y18 (1:0.6:0.6)	22.0 ± 0.4	899 ± 2	64.01 ± 0.8	12.7 ± 0.2
T2	PM6:Y6:Y11 (1:0.6:0.6)	26.1 ± 0.4	836 ± 2	69.07 ± 0.1	15.1 ± 0.3
Q1	PM6:Y6:Y11:Y16 (1:0.4:0.4:0.4)	23.0 ± 0.6	880 ± 10	58.7 ± 0.03	11.9 ± 0.9

Q2	PM6:Y6:Y11:Y18 (1:0.4:0.4:0.4)	26.3 ± 0.5	857 ± 7	67.7 ± 1.5	15.3 ± 0.5
Q3	PM6:Y1:Y16:Y18 (1:0.4:0.4:0.4)	19.6 ± 0.3	920 ± 4	58.2 ± 1.4	10.5 ± 0.3
P1	PM6:Y1:Y6:Y11:Y16 (1:0.3:0.3:0.3:0.3)	22.0 ± 0.1	902 ± 2	60.3 ± 0.5	11.98 ± 0.1
P2	PM6:Y6:Y11:Y16:Y18 (1:0.3:0.3:0.3:0.3)	22.9 ± 0.5	817 ± 31	35.8 ± 0.9	6.84 ± 2.2
H1	PM6:Y1:Y6:Y11:Y16:Y18 (1:0.24:0.24:0.24:0.24:0.24)	25.4 ± 0.3	880 ± 5	62.0 ± 0.01	13.7 ± 0.4
H2	PM6:Y1:Y6:Y11:Y16:Y18 (1:0.1:0.4:0.2:0.1:0.2)	24.7 ± 0.3	871 ± 4	66.1 ± 2.4	14.21 ± 0.6
H3	PM6: Y1:Y6:Y11:Y16:Y18 (1:0.2:0.8:0.6:0.1:0.2)	22.6 ± 0.3	853 ± 9	67.5 ± 1.7	13.02 ± 0.6
H4	PM6: Y1:Y6:Y11:Y16:Y18 (1:0.1:0.8:0.15:0.15:0.6)	23.6 ± 0.9	855 ± 5	67.5 ± 2.7	13.63 ± 1

3. All acceptors were mixed with equal weight ratios in this study. Did the authors evaluate other weight ratios? Or say, whether this generic strategy depends on the weight ratio in the acceptor mixture as well?

We chose to work with equal weight ratios because our hypothesis was that the entropy of mixing, which is maximized for equimolar mixtures, influences the device stability. Note that all Y-series (ITIC) acceptors have a similar molecular weight, and hence weight and molar fractions are similar.

Also, the composition-device efficiency diagram shown in Fig. S6 reveals that hexanary devices with an equal-weight ratio are located close to a local maximum in terms of device efficiency (see SI Fig. 6). Therefore, we argue that devices with an equal weight ratio are suitable model systems to study the impact of mixing of several acceptors on the device stability.

Given the vast number of possible acceptor combinations, evaluating a wider range of multi-component devices becomes prohibitively labor intensive. It can be anticipated that this work inspires follow-up studies that use high-throughput robot-based device fabrication or the characterization of devices with a composition gradient (Martinez, X., et al., Predicting the photocurrent–composition dependence in organic solar cells, Energy & Environmental Science 14(2): 986-994, 2021) to identify blend compositions that offer both optimal device performance and stability among multi-component devices.

This part is added in the device part of the results section (page #11, para 1)

“Organic solar cells based on binary, ternary, quaternary, pentanary and hexanary blends with a wide range of acceptor compositions (selected from Y1, Y6, Y11 Y16 and Y18 (see Figure 1 for chemical structures) were fabricated with

a conventional device architecture (see Methods section for details, *J-V* curves of Q1, B1 and H1 based devices in Fig. S5a and summary of device parameters in Table 1 and Table S4). Also, Fig. S6 reveals that hexanary devices with an equal weight ratio of acceptors (H1) feature a performance that is comparable to binary devices (e.g. B1, cf. Table 1) and on a par with other hexanary devices that comprise different donor:acceptor ratios (cf. Table S4). Therefore, we chose to compare devices comprising equal weight fractions of the acceptors in the remainder of this study.”

We have also expanded the discussion section:

“While the impact of mixing on the thermal stability is applicable to different types of NFA mixtures (e.g. Y- and ITIC-type acceptors), the optimal number of acceptors in a mixture may vary for different types of NFAs, thus providing an additional parameter that can be adjusted, leading to further gains in stability and device performance. Given the vast number of possible acceptor combinations, evaluating a wider range of multi-component devices becomes prohibitively labor intensive. It can be anticipated that this work inspires follow-up studies that use high-throughput robot-based device fabrication (Du, X., & Lüer, L., et al., Elucidating the Full Potential of OPV Materials Utilizing a High-Throughput Robot-Based Platform and Machine Learning, *Joule* 5(2): 495-506, 2021) or the characterization of devices with a composition gradient (Martinez, X., et al., Predicting the photocurrent–composition dependence in organic solar cells, *Energy & Environmental Science* 14(2): 986-994, 2021) to identify blend compositions that offer both optimal device performance and stability among multi-component devices.”

4. On page 13, the authors mentioned that this multiple-component strategy has been proved with another type of acceptor, ITIC derivatives, where a similar thermal stability improvement has been observed. But the device performances summarized in Table S4 also showed that in the ITIC system, the performance of hexanary-3 device was poorer than all binary devices. Is this efficiency loss a potential issue that should be taken into consideration when using the multiple-acceptors strategy?

Though the highest efficiency binary devices based on ITIC type acceptors (B8) have a one-third higher efficiency compared with the hexanary devices (H6), the performance of B8 devices decreases rapidly during the first few days of annealing. In contrast, the H6 devices lose only 5% of their efficiency after annealing for 25 days. The initial efficiency of the hexanary devices should be the basis of comparison only if the T_{80} (time required to reach 80% of initial efficiency) of both binary and hexanary devices is similar. If T_{80} of hexanary devices is significantly longer than the value for the respective binary devices, the initial efficiency of the hexanary device need not be a potential issue. After a relatively short period of annealing (e.g. 5 days at 130 °C in case of H6 and B8) hexanary devices will display a superior performance compared with binary devices.

We added this discussion in the device part of the results section (page 14, para 1)

“To understand if the use of multi-component mixtures can be extended to other types of acceptors, similar annealing studies were conducted for hexanary devices based on ITIC derivatives (H6; for device parameters see Table S5 and Fig. S9). In contrast to hexanary devices based on Y6-type acceptors, the initial efficiency of H6 devices is one-third lower than corresponding binary devices (B6-B8; Table S5). The lesser complementary absorption among the ITIC derivatives and the donor polymer PM6 (Fig. S10) is the likely cause for the lower J_{sc} of H6 devices compared with the binary devices B6-B8 (Table S5). Further, the ITIC derivatives that make up H6 devices feature an up to 0.41 eV difference in E_{A_A} (see Table S1), which results in a relatively high degree of energetic disorder that limits the device performance. Gratifyingly, H6 devices maintain their performance when annealed for 23 days at 130 °C, while corresponding binary devices display a significant decrease in device efficiency (Fig. S11). As a result, after only 5 days of annealing at 130 °C the efficiency of H6 devices exceeds that of, e.g., B8 devices. Evidently, mixing of a large number of acceptors is a strategy that can be extended to other classes of NFAs (Fig. S11).”

Fig. S2. Absorbance spectra of donor and acceptors used in H1-4 (a), H5 (b) and H6 (c)

5. Devices made by doctor-blade casting in ambient conditions achieved even higher power conversion efficiency. Did the authors have any explanation for the better performance of the doctor-blade casting device?

Different film deposition techniques usually tend to result in a different nanostructure. To investigate the differences in nanostructure of spin-coated and doctor-bladed active layers of hexanary blends, we have recorded AFM images, provided in SI Figure S7. For the studied films, AFM indicates that different processing techniques results in films with a comparable surface topography. Based on this result, we have revised our claim that blade-coating results in a higher device performance. A closer look at the error bars in Table 1 for the device parameters of devices with a spin-coated or doctor-bladed active layer suggest that the difference in the mean values ($17.1 \pm 0.4 \%$ vs $17.6 \pm 0.4 \%$) is not statistically significant, i.e. the two types of devices behave the same. We have modified the relevant section in the results part:

“Doctor blading of the H5 active layer in air resulted in a PCE = 17.6 %, which, considering the error in J_{SC} , is statistically similar to the performance of spin-coated devices (see Table 1). The invariance of the device efficiency is consistent with AFM images of spin-coated and doctor-bladed films, which reveal a comparable surface topography (Fig. S7). It can be concluded that OPV blends comprising a mixture of several acceptors can be processed with high-throughput printing techniques (Table 1 and Fig. S5b).”

Fig. S3. Atomic force microscopy (AFM) phase and height images of (a, b) doctor-bladed and (c, d) spin-coated H5 active layer films.

The experimental details of the film preparation and the film characterization can be found in the methods section.

Reviewer #2 (Remarks to the Author):

Authors demonstrate that hexanary blends (5 acceptors + 1 donor) improve the thermal stability of organic PV cells. Indeed, one major cause of degradation is crystallization of the small molecules NFA, which the increase of entropy provided by blending multiple acceptors together forbids. The authors' claim is witnessed in detail by several techniques like TDCF, GIWAXS and FSC.

The manuscript is well written and convincing, and the report is original and sound. A few details should be considered to improve the manuscript:

1. what was the area of the devices under test?

The active area of the devices was 0.1 cm²; mentioned in relevant figure legends in the following parts of the main text.

“Table 1. Mean and standard deviation of photovoltaic device parameters for 6 as-cast devices on the same substrate with the following active layer compositions: Binary (B1): PM6:Y6 (1:1.2); Quaternary (Q1): PM6:Y6:Y11:Y16 (1:0.4:0.4:0.4); Hexanary-1 (H1): PM6:Y1:Y6:Y11:Y16:Y18 (1:0.24:0.24:0.24:0.24:0.24) and Hexanary-5 (H5): PM6:Y6:Y7:Y18:N3:BTP-eC9 (1:0.24:0.24:0.24:0.24:0.24). The donor:acceptor ratios are given as weight ratios. *Fabricated by doctor blading in air. **The active area of each pixel is 0.1 cm².**”

“Fig. 1. a) FF, b) PCE of H1, Q1 and B1 devices recorded before and after annealing the active layer up to 23 days at 130 °C; data points represent the mean and standard deviation of measurements done for 6 pixels on the same substrate. **The active area of each pixel is 0.1 cm².** Time delayed collection field measurements with the total extracted charge Q_{tot} (dotted lines) as a function of the applied pre-bias overlaid with the respective J-V curves (solid lines) for the fresh and aged B1 (c) and H1 (d) devices.”

“Fig. 2. T_{95} , is the time for binary (B1) and hexanary devices (H1) to reach 95% of their initial performances. T_{95} with three different active layer thickness. **The active area of each pixel is 0.1 cm².**”

2. we understand that increasing the entropy of mixing of a mixture favors the liquid (or glassy state - which is a frozen liquid). However, authors claim that the composition under study has "unprecedented ability to form a molecular glass". It is clear that the hexanary blend was unprecedented, but revealing the interest of glass formers to improve the stability of PSCs was not unprecedented, as reported in the review 'Revisiting the Optimal Nano-Morphology: Towards Amorphous Organic Photovoltaics' (DOI: 10.1002/tcr.201800158).

The reviewer misunderstood this sentence in the introduction. The “unprecedented ability to form a molecular glass” refers to a recent study where we have investigated glass formulation of mixtures comprising up to eight

different perylene derivatives (Sci. Adv. 7, eabi4659, 2021). We discuss this previous work to provide insight into why we chose to study multi-component acceptor mixtures in the context of OPV. The reviewer is of course absolutely correct with stating that glass formation of OPV blends is a widely studied approach.

We appreciate the suggestion for additional literature, which we now include as a reference in the introduction:

“One prominent way to impede long-range mass transport is the use of glassy BHJ blend films with a glass transition temperature T_g far above device operating conditions (de Zerio, A.D., et al., Glass Forming Acceptor Alloys for Highly Efficient and Thermally Stable Ternary Organic Solar Cells, *Advanced Energy Materials* 8, 1702741, 2018; Nunzi, J.-M., et al., Revisiting the Optimal Nano-Morphology: Towards Amorphous Organic Photovoltaics, *The Chemical Record* 19, 1028-1038, 2019; Peng, Z., et al., A materials physics perspective on structure–processing–function relations in blends of organic semiconductors. *Nature Reviews Materials*, 2023.)”

Reviewer #3 (Remarks to the Author):

In this manuscript, D. Baran and co-authors constructed a hexanary organic photovoltaic and conducted a systematic study on the thermal stability of the device. The authors demonstrated that the hexanary blend has better thermal stability, and tried to explain the mechanism behind it. This manuscript is data-rich but not innovative.

We thank the reviewer for the critical comments. In our view the innovative point of our manuscript is the introduction of a novel rationale for improving the thermal stability of OPV blends, which is based on mixing of significantly more components than what has been considered up to now. We anticipate that this insight will spur further studies into multi/component mixtures, possibly even exceeding 6 components.

However, if it wants to be published in Nature Communications, the reviewer consider at least that the following issues need to be discussed:

1. Page 6, line 139, the authors consider that the melting enthalpy of the material can be equivalent to the total amount of crystallization. What are the prerequisites for comparing different acceptor films?

That is a very valid criticism. The acceptor mixtures either contain fewer and smaller single-acceptor crystallites, or they feature aggregates of several different acceptors, an argument that we also follow up on in the discussion section. We have rewritten the relevant part of the results sections:

“The addition of two more acceptors resulted in even lower values of $T_m = 256$ °C and $\Delta H_m = 10$ J g⁻¹ in case of the pentanary mixture Y1:Y6:Y11:Y16:Y18 (Table S2). A similar trend is observed for two other pentanary acceptor mixtures (see Fig. S2). The absence of melting endotherms above 280 °C suggests that crystallization of single-acceptor crystallites is strongly suppressed (note that all Y6 polymorphs melt just below 300 °C (Gutierrez-Fernandez. E., Y6 Organic Thin-Film Transistors with Electron Mobilities of 2.4 cm² V⁻¹ s⁻¹ via Microstructural Tuning, Advanced Science 9(1): 2104977, 2022)). We argue that mixing increases the entropy of the liquid state in case of ternary and pentanary mixtures, which reduces the driving force for crystallization (cf. introduction). The remaining broad endotherm with a lower T_m and ΔH_m may arise because residual single-acceptor crystallites are still present, which are however smaller in size. Alternatively, aggregates of several acceptors may have formed, as discussed below.”

2. Page 12, The authors claim that blade coating can achieve better device performance than spin coating, which indicates that the processing method has impacts on the morphology and performance of devices.

We thank the reviewer for this comment. Reviewer 1 had also commented on the same part. We have carried out AFM to investigate if spin coating and doctor blading result in very different surface topographies. Interestingly, films

prepared by these two methods result in very similar topographies (see Figure S7, also included above). Based on this result, we revise our claim that blade coating results in a better device performance and instead argue that the two processing techniques result in devices with a comparable performance. This is also in agreement with the errors in Table 1, based on which the device efficiencies are not statistically different ($17.1 \pm 0.4 \%$ vs $17.6 \pm 0.4 \%$). We have modified the discussion the main text as described above.

However, when discussing the film thickness-dependent stability, the authors ignore the morphology difference caused by different spin-coating speeds and solution concentrations, and the resulting impact on stability. Please add this discussion.

Devices with a thickness of approx. 40, 100 and >300 nm were compared. To fabricate devices active layers were prepared by different processing techniques, i.e. spin coating inside glovebox and doctor blade in air. While changes in processing conditions can strongly impact the nanostructure of BHJ blends, we note that spin-coating and doctor-blading nevertheless result in hexanary blend films with a similar surface topography (Figure S7). The differences in processing conditions may explain the observed variations in T_{95} . For hexanary devices, however, we observe a significantly higher T_{95} independent of film thickness, which is in agreement with a nanostructure that is less sensitive to changes in processing conditions. The text is highlighted in the results section on page 17 of the main text.

While B2 devices with thin active layers (B2_1) retain their photovoltaic performance upon annealing, in case of thick B2 (B2_2) devices we observe an 86 % drop in PCE upon annealing at 130 °C for 14 days, predominantly due to the loss in J_{sc} and FF. The slightly higher T_{95} of B2_3 could be due to the different nanostructure that results from different processing conditions. Instead, the stability of hexanary devices appears to be less dependent on the processing conditions.

3. Page 13, For binary devices, the total extracted charge shows the same trend with bias voltage in fresh and aged devices, and the same phenomenon was also observed in hexanary devices (Fig. 4c and Fig. 4d). But what confuses the reviewers is the author's claim that "the loss in FF of B1 based devices is due to a decrease in the charge transport properties obtained upon ageing of the active layer", Please explain in detail. In addition, Fig. 4 and Fig. S9 suggest that H1, H2 exhibit bias-dependent charge extraction, why the authors claim that "Devices based on H1, H2, and B1 blends do not exhibit any changes in the bias dependence of charge generation" in page 14?

We thank the referee for pointing out the possible confusion due to the sentence structure. We meant that the charge generation behavior in H1, H5 and B1 devices is not affected by their aging. The text is highlighted in the results section on page 14 of the main text.

“We carried out time delayed collection field (TDCF) experiments to understand how ageing of the active layer influences charge separation and recombination process and thus the device parameters. Charge generation in case of H1, H5 and B1 devices displays a slight field dependence but is not affected by ageing of the respective devices (Fig. S12) and thus cannot explain the observed loss in FF, which is most pronounced in case of binary devices (see Fig. S8). However, devices based on aged B2 blends showed a clear change in the bias dependence of the charge generation, which is consistent with the loss in J_{sc} observed upon annealing (Fig. S8a). Interestingly, both binary (B1, B2) and hexanary (H1, H5) devices feature a similar recombination constant prior to and after aging (Fig. 4 c-d). The similar charge generation and extraction properties of as-cast and aged devices suggests that the loss in FF of B1 based devices is due to a higher series resistance in the aged active layer compared with the respective as-cast B1 active layer (Jao. M-H., et al., Achieving a high fill factor for organic solar cells, *Journal of Materials Chemistry A* 4(16): 5784-5801, 2016) (Fig. S13). In addition, Wöpke et al. have assigned the increase in trap density upon annealing to the crystallization of PM6 (Wöpke. C., et al., Traps and transport resistance are the next frontiers for stable non-fullerene acceptor solar cells, *Nature Communications* 13(1): 3786, 2022).”

4. The thermal aged film (Q, H1) shows larger CL values of acceptor and PM6 than fresh film, however, the aged and fresh B1 film shows similar CL values of acceptor and PM6 (Table S3). This seems to indicate that B1 has better thermal stability, so why would the device based on B1 film be less thermally stable?

GIWAXS mostly provides information about ordered phases but not mixed donor:acceptor regions, which play an important role in charge generation, as discussed by a number of recent papers Collins. B.A., et al., Absolute measurement of domain composition and nanoscale size distribution explains performance in PTB7:PC₇₁BM solar cells, *Advanced Energy Materials* 3(1): 65-74, 2013; Mukherjee. S., et al., Significance of average domain purity and mixed domains on the photovoltaic performance of high efficiency solution-processed small-molecule BHJ solar cells, *Advanced Energy Materials* 5(21): 1500877, 2015). Since the change in loss in device parameters does not correlate with the coherence length (CL), we argue that other aspects impact the thermal stability. In the discussion section we specifically discuss how differences in the makeup of the mixed phase, driven by differences in acceptor solubility in the polymer at thermal equilibrium, can account for the observed improvement in thermal stability in case of multi-component (hexanary) devices.

Accordingly, we have expanded the discussion of the GIWAXS results on page 15, as described below.

“We argue that (the degree of change in) the size of ordered acceptor domains is not a good metric for judging the performance and thermal stability of devices that comprise mixtures of Y-series acceptors. Instead, changes that are not captured by GIWAXS likely cause the decrease in FF of B1 devices upon thermal annealing, such as changes in the composition of the disordered phase(s).”

5. Page 16, The authors claim that “the superior thermal ability of hexanary blends also persists in the case of 117 nm thick films... may facilitate the fabrication of thermally stable devices via high-throughput processing techniques.” However, the communities generally consider that 300nm is the minimum thickness requirement for industrial production, hence it is recommended to supplement the stability of 300nm devices or modify the language.

We thank the reviewer for this comment, we agree with the reviewer that the films as thick as 300 nm would provide flexibility to optimize the photon harvest of the active layer for high throughput printing techniques like roll-to-roll coating. But, we would like to mention that the photo-active-layer as thick as 70 nm could also be printed with high-throughput techniques (Chaturvedi. N., et al., All slot-die coated non-fullerene organic solar cells with PCE 11%, *Advanced Functional Materials* 31(14): 2009996, 2021).

Nevertheless, thermal stability of the devices with an active layer thickness of about 300 nm were fabricated by doctor-blading. Stability studies show the 390 nm thick hexanary devices do not experience any decrease in efficiency over a period of 14 days. Instead, under the same conditions the efficiency of binary devices with a thickness of 310 nm has decreased by about one-fourth. We have added the following discussion to the main text:

“In a further set of experiments, we examined the impact of the active-layer thickness on the thermal stability of H1 and B2 devices with three different active layer thicknesses each (see Table S7 for thickness values). Among these 6 different devices, H1_3 and B2_3 devices were doctor-bladed in air, and the other four types of devices were spin-coated in a glovebox (see Table S8 for device performances). Both, binary and hexanary devices with thicker films (H1_3 and B2_3) suffer from a lower V_{oc} compared to the respective thinner active layer film devices. This is likely due to the higher charge recombination in the thicker active layer films owing to the limited exciton diffusion lengths of the NFAs (Firdaus. Y., Le Corre. V.M., et al., Long-range exciton diffusion in molecular non-fullerene acceptors, *Nature Communications* 11(1): 5220, 2020; Markina. A., et al., Chemical design rules for non-fullerene acceptors in organic solar cells, *Advanced Energy Materials* 11(44): 2102363 2021). In addition, B2_3 devices suffer from a lower J_{sc} compared with the thinner binary devices (B2_1 and B2_2), whereas H1_2 and H1_3 devices feature similar J_{sc} values.

All devices were annealed at 130 °C for 14 days in the dark and under an inert atmosphere (see Methods section for details). The thermal stability of B2 devices showed a significant dependence on the active layer thickness (Fig. 5; more details can be found in Table S8 and Fig. S15-16). While B2 devices with thin active layers (B2_1) retain their photovoltaic performance upon annealing, in case of thick B2 (B2_2) devices we observe an 86 % drop in PCE upon annealing at 130 °C for 14 days, predominantly due to the loss in J_{sc} and FF. The poor J_{sc} and FF of the as-casted B2_3 devices likely masked a further drop in photovoltaic performance upon annealing, which is reflected in the slightly longer T_{95} .”

The following data were added to the manuscript and SI.

Figure 3. **Impact of active layer thickness on thermal stability.** The time T_{95} for binary (B2) and hexanary devices (H1) to reach 95% of their initial PCE upon annealing at 130 °C, measured for different active layer thicknesses (for H1 devices T_{95} had not been reached at the end of the 14-day stability test). The active area of each pixel is 0.1 cm².

Table S2. The thickness of the active layer films referred to in figure 5 of the main text.

Sample no.	Device name	Thickness, nm
01	H1_1	44
02	H1_2	117
03	H1_3	390
04	B2_1	38
05	B2_2	94
06	B2_3	310

Table S3. Mean and standard deviation of photovoltaic device parameters for 6 devices on the same substrate. H1 and B2 of fresh and aged devices with different active layer thickness. The donor:acceptor ratios are weight ratios.

		as prepared				aged at 130 °C for 336 h			
Film Thickness (nm)		J_{sc} (mA cm ⁻²)	V_{oc} (V)	FF (%)	PCE (%)	J_{sc} (mA cm ⁻²)	V_{oc} (V)	FF (%)	PCE (%)
H1	43	13.8 ± 0.4	892 ± 1	62.2 ± 0.01	7.7 ± 0.3	15.8 ± 0.3	834 ± 8	60.5 ± 0.03	7.9 ± 0.3
	112	21.8 ± 0.3	895 ± 1	59.2 ± 0.01	11.6 ± 0.3	22.2 ± 0.5	859 ± 2	56.8 ± 0.02	10.8 ± 0.5
	390	21.3 ± 0.6	703 ± 6	51.3 ± 0.01	7.7 ± 0.2	22.9 ± 0.1	746 ± 0	42.5 ± 0.00	7.3 ± 0.0
B2	40	15.2 ± 0.5	864 ± 2	68.8 ± 0.01	9.1 ± 0.3	15.5 ± 0.4	784 ± 20	61.1 ± 0.03	7.5 ± 0.3
	90	25.3 ± 0.4	872 ± 2	65.9 ± 0.01	14.5 ± 0.3	5.8 ± 0.4	748 ± 20	46.1 ± 0.03	2.0 ± 0.3
	310	20.2 ± 0.7	770 ± 3	53.8 ± 0.01	8.4 ± 0.3	20.5 ± 0.1	633 ± 1	49.8 ± 0.00	6.5 ± 0.0

Fig. S4. Representative J-V curves of H1 (a) and B2 (b) with different active layer thickness.

Fig. S5. a) J_{sc} , b) V_{oc} , c) FF and d) PCE of H1, and B2 devices with different active layer thickness recorded at regular intervals by annealing half devices up to 336 h (14 days) at 130 °C; data points represent the mean and standard deviation of measurements done for up to 6 pixels on the same substrate.

6. Page 11, In order to visually compare 2D GIWAXS images, the intensity scale should be added.

We have added the intensity scale to the 2D GIWAXS images. The changes can be seen in the fig 3. in the main text and fig. S14 in SI.

Fig. 4. GIWAXS patterns of thin films of Y6 (a), Y6:Y11:Y16 (b), Y1:Y6:Y11:Y16:Y18 (c), PM6:Y6 (d), PM6:Y6:Y11:Y16 (e) and PM6:Y1:Y6:Y11:Y16:Y18 (f) spin-coated from chloroform and annealed at 120 °C for 1 min.

Fig. S6. GIWAXS patterns of PM6:Y6 (a), PM6:Y6:Y11:Y16 (b) and PM6:Y1:Y6:Y11:Y16:Y18 (c) after spin-coated from chloroform and annealed at 120 °C for 1 min, followed by ageing for 120 h at 130 °C. The thickness of all the GIWAXS conducted films can be found in table S5.

7. Page 14, line 279, the coherence length summarized in Table 1 or Table S3?

We have corrected for the typo. The reference is corrected as Table S3 in the page 15 as shown below.

The acceptor domains of B1 blend films do not change upon annealing, as indicated by a similar coherence length of ordered domains of about 9 nm (see Table S3).

REVIEWERS' COMMENTS

Reviewer #1 (Remarks to the Author):

The authors have carefully addressed all the questions raised by the reviewers, providing comprehensive explanations and supplementary data where necessary. Based on the thorough revisions made, I believe the manuscript is now suitable for publication in Nature Communications.

Reviewer #2 (Remarks to the Author):

thank you for clarifying your point in the manuscript

Reviewer #3 (Remarks to the Author):

Improving the thermal stability of OPV blends, especially thick blends, is really important. In this manuscript, the authors' finding is vital for further designing multi-component systems to enhance the thermal stability of OPVs. The revised manuscript has well addressed the concerns raised by the reviewers, and I recommend this work be considered for publication in Nature Communication.